# Improving Motivation and Learning Experience with a Virtual Tour of an Assembly Line to Learn about Productivity

Mónica Hernández-Campos [1,*], Luis Carlos Guzmán-Arias [2], José Fabián Aguilar-Cordero [1], Edgar Rojas-Muñoz [3], Ronald Leandro-Elizondo [4] and Yuen C. Law [5,*]

1 Center for Academic Development, Instituto Tecnológico de Costa Rica, Cartago 159-7050, Costa Rica; jfaguilar@itcr.ac.cr
2 Tec Digital, Instituto Tecnológico de Costa Rica, Cartago 159-7050, Costa Rica; luguzman@itcr.ac.cr
3 School of Performance, Visualization & Fine Arts, Texas A&M University, College Station, TX 77843-3237, USA; ed.rojas@exchange.tamu.edu
4 School of Business Administration, Instituto Tecnológico de Costa Rica, Cartago 159-7050, Costa Rica; rleandro@itcr.ac.cr
5 School of Computer Engineering, Instituto Tecnológico de Costa Rica, Cartago 159-7050, Costa Rica
* Correspondence: mohernandez@itcr.ac.cr (M.H.-C.); ylaw@itcr.ac.cr (Y.C.L.)

**Abstract:** We propose the use of a Virtual Tour to substitute in-person visits to a manufacturing plant for a lecture on Enterprise Productivity at the School of Business Administration at our University. Traditionally, during this lecture, students are required to visit a production site to observe its process and apply their knowledge in a real-life scenario. However, finding businesses that are willing to participate and offer the right learning conditions has become a challenge. This situation is now worse due to the COVID-19 pandemic. In this paper, we present a prototype of a virtual tour of an assembly line in a simulated environment, where students can explore and learn about the manufacturing process of car seats. We performed a mixed method user study, with quantitative and qualitative data, to determine whether the application can help learn the intended concepts and improve the learning experience and motivation of students. Results show that the use of the virtual tour application increased motivation in learning.

**Keywords:** immersive technologies; virtual tour; motivation; user experience; business production teaching; mixed study

## 1. Introduction

The world has moved toward a global business transformation associated with the Fourth Industrial Revolution, or Industry 4.0. Industry 4.0 refers to how emerging technologies, like the Cloud, robotics and cobotics, the Internet of Things, automation, mobility, etc., are disrupting how products are being produced and delivered to the markets [1]. Indeed, this new era has been characterized by a more complex, interconnected, and holistic understanding of manufacturing [2,3]. Given such a scenario, it is a valuable opportunity for higher education institutions to use those technologies to improve teaching through innovative pedagogical and technology-assisted strategies, facilitating access to quality education in facing current global problems and redesigning tomorrow's world [4]. Moreover, the emergency created by SARS-CoV-2 has resulted in the necessity to generate learning spaces in higher education that allow students to participate in practical courses or laboratories that are like those they would have in normal conditions.

Mixed reality immersive applications could replace or complement real-life experiences, which are fundamental in achieving desired learning outcomes. Furthermore, these applications offer the advantage that students can relive the experiences at their own rhythm and repeat them as many times as required [5,6]. In Business Administration academic programs, visiting a production facility to observe their manufacturing processes

and apply the acquired knowledge to a real-life situation is a valuable pedagogical tool. However, due to various reasons, it has become more difficult to find companies willing to host such visits and that will also present the adequate conditions to offer students the correct learning experience they need. This issue has been further worsened by the pandemic. In this study, a Virtual Tour (VT) application was designed to substitute in-person visits to production facilities. The purpose of this study was to evaluate the implementation of this virtual tour in the context of a lecture on learning, motivation, and user experience through a mixed research design.

The contributions of this paper are as follows: first, in contrast to others, we propose a VT application as a tool to provide a learning experience specially designed to meet the requirements of a course curriculum; second, we revised and updated the 'traditional' learning methodology, class materials, activities, and evaluations according to current principles to integrate active learning and immersive technologies; and third, we performed a user study to compare the use of the virtual tour application, versus the updated 'traditional' (without interactivity) material. We present qualitative and quantitative results.

The remainder of this article is structured as follows: we review the related work in the next section. Then we present the material and methods in Section 3. The description of the quantitative and qualitative studies is presented in Section 4, and their respective results are in Section 5. We discuss our findings in Section 6, and finally, Section 7 presents conclusions and future work.

## 2. Related Work

### 2.1. Virtual Tours for Education and Training

A virtual tour (VT) is generally described as a digital representation or simulation of a location that can be presented as a series of images or videos [7]. Other forms of VT take advantage of immersive technologies and virtual reality to recreate the location in a 3D digital environment and to allow visitors to explore it interactively. Previous studies reported a positive effect on learning through the use of immersive technologies that emulate the manufacturing processes in educational contexts [8–11]. Other studies have also found positive results in learners' mood and motivation when utilizing virtual reality (VR) [12–14]. For instance, Su et al. [15] conducted an experiment to investigate whether VR immersive technology enhances mathematics geometry learning and motivation. Their research hypothesis was that learning motivation is a crucial factor that affects learning, and motivated students show better learning outcomes. In their study, the experimental group used a VR immersive learning mathematics geometry system. The results showed that the experimental group exhibited better learning outcomes compared to the control condition, suggesting that virtual technology is an appealing tool for learning.

Hamilton et al. [16] conducted a systematic literature review focusing on learning outcomes, intervention characteristics, and assessment measures associated with the use of immersive virtual reality (I-VR) in education. The review revealed that the research in this area has been limited, but most studies have shown a significant advantage of utilizing I-VR in education. The review emphasizes the importance of a rigorous methodological approach that includes appropriate assessment measures, intervention characteristics, and learning outcomes in order to fully understand the potential of I-VR as a pedagogical method.

Despite the positive findings on learning and motivation associated with immersive technologies, systematic literature reviews have identified certain limitations in studies focused on virtual applications in educational contexts. Matovu et al. [14] conducted a systematic literature review and found that learning theories did not play a significant role in the design, implementation, and evaluation of IVR-based learning. These findings align with those of Radianti et al. [17] who discovered that the evaluation of educational VR applications primarily focused on usability rather than learning outcomes. Furthermore, immersive VR has been predominantly used in experimental and developmental work rather than being regularly applied in actual teaching. In another systematic literature review, Won et al. [18] found that pedagogical features of immersive technologies in educational

contexts lack detailed explanations on how they are integrated within the instructional design. Given the high costs and time involved in designing educational immersive technologies, these authors recommend that future researchers consider objectives, content, and environmental constraints to develop effective and practical technological solutions.

Based on the findings from previous studies, our research aims to develop a virtual tour of a manufacturing process in a real educational context. In addition, the design of the virtual tour takes into account the "cognitive load theory" to enhance learning. Finally, the virtual tour is integrated within a real educational context, considering aspects such as purpose, content, pedagogic mediation, and assessment.

### 2.2. Cognitive Theory and Immersive Technologies

The evidence indicates that the use of simulations and immersive technologies for education is effective for the development of complex abilities [19]. A theory that supports the design of these strategies and learning resources is the theory of cognitive load [20], which is derived from the information processing model of Atkinson and Shiffrin [21] and serves as a background for some practices in education [22]. This instructional approach considers human cognitive architecture for facilitating learning in educational settings, and takes into account the limited capacity of working memory, the organization of long-term memory, and the interaction between both systems [20]. The design of the virtual tour of this research considers the main principles of cognitive load theory [22]:

- Split attention effect and spatial contiguity principle: The placement of visual elements and their text labels should follow the spatial contiguity to avoid split user attention and two separate stimuli sources. The drawback of split attention is that it will demand more working memory resources and will reduce the learning of the content.
- Modality effect: The design of learning resources should consider short auditory descriptions and visual information in order to not saturate any information path (visual or auditory).
- Redundancy effect and coherence principle: The learning object should not include redundant information. Only include the more relevant written information.
- Signaling principle: Learning objects are more effective when including elements to cue their essential parts.

Another topic to consider is the principle of multimedia, which explains that different people learn better when text and images are presented to them together. This same principle suggests that the use of interactive animations is very effective while learning procedures, especially in beginners [23]. The use of multi-modal elements makes the process of learning easier and more effective, promoting creativity and motivation, while also improving comprehension [24].

## 3. Materials and Methods

### 3.1. Learning Methodology

As mentioned, it is important that learning tools such as the VT application we are proposing are accompanied by careful planning of the learning experience. Before designing the Virtual Tour, the planning of the lecture was accomplished following the principles of instructional design, psychology, and learning [25]. In this way, the virtual tour was used as a learning tool inserted in a methodology in which the objectives, contents, mediation strategies, educational resources, and assessment were congruent. It was expected from this class that students were able to evaluate the productive process through the use of some work-study techniques, for the reduction of waste. The contents were work-study techniques, and waste reduction in manufacturing processes. The final lecture was structured as follows:

- Slide presentation with the description of the content. Images and audio were available for participants.
- Review of a production process within a fictitious production facility.

- Active learning strategy where students analyze the production facility.
- Assessment.

### 3.2. Virtual Tour

A VT application was developed using the Unity 3D game engine. The virtual factory is presented as a 3D space that the user can explore in a first-person perspective; that is, the view appears as if the camera was positioned on the eyes of the character the user is controlling, as can be seen in Figure 1. We also use typical gaming controls for character movement: the keys W, A, S, and D for forward, left, back, and right, respectively; the mouse to change the view direction; and left-click for actions. Table A4 of Appendix D details the technologies used in the development of the VT.

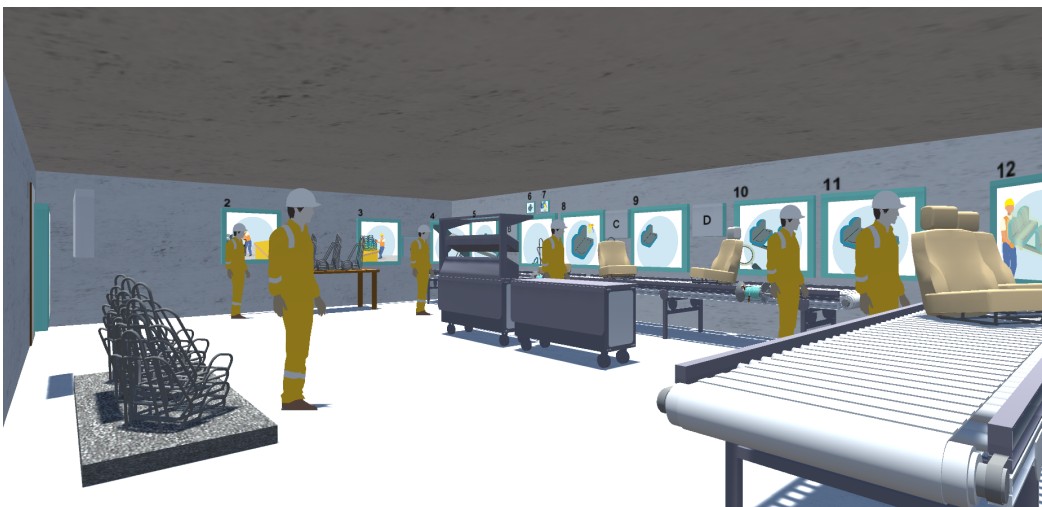

**Figure 1.** First-person view of the virtual manufacturing plant, as presented in the Virtual Tour.

The VT shows a simplified version of the production process of car seats. As this virtual tour was meant to illustrate different concepts in production, such as storage, delay, and transport, the process's steps and stations were selected to at least show one of these. The VT was divided into different stages distributed around a room; each of these stations showed one stage of the production, from transporting the metal frames to the workstations, through inspection and ending in packing and storage. Process stages were numbered and the workstations were labeled with letters to allow easy navigation through the steps, but users were able to freely explore the stations in whatever order they preferred, although following the sequential order is encouraged by the layout.

Each station has a virtual screen that illustrates the stage in the assembly process with an animation. A voice recording explains each step individually. The animations and the accompanying audio can be played and paused individually by clicking on the screen. If a new screen is clicked while another animation is playing, the previous one will automatically stop before the new one starts. Animations and audio tracks can be played as many times as needed.

## 4. Evaluation

### 4.1. Method

This study was developed through a descriptive and convergent mixed method design QUAN/QUAL in which quantitative and qualitative data are collected concurrently, but separately [26]. The purpose of employing this mixed design is to compare and analyze data in order to enhance our understanding of the phenomenon. By doing so, the researchers aim to gain a more comprehensive understanding of students' experiences in utilizing the VT as a pedagogical resource and to assess the impact of the VT in comparison to traditional teaching materials on content comprehension, usability, and motivation. Thus,

a study design with a control and an experimental group was chosen. Both groups studied the same class material, which was redesigned to apply an active learning methodology. The difference between both groups is the main learning activity. During this activity, the experimental group used the Virtual Tour, while the control group used traditional media, i.e., a slide presentation.

Research Question 1 (RQ1): Can the use of a Virtual Tour as a pedagogic resource improve students' understanding and motivation toward the class content in comparison to the control group?

Research Question 2 (RQ2): Are the scores of the user experience superior in the experiment group in comparison to the control group?

Research Question 3 (RQ3): What was the opinion of participants of the experimental condition regarding the usability and contributions of the VT?

### 4.2. Quantitative Study

According to previous research, the use of immersive technologies improves the quality of the learning experience. We focus on the aspects of motivation, user experience, and conceptual understanding as measures to determine the overall learning experience. We formulate three hypotheses as follows:

**Hypothesis 1.** *The students will find the user experience of the VT better than the user experience of the traditional material. We believe that the students will find the virtual tour application to be more innovative and engaging compared to the traditional material.*

**Hypothesis 2.** *The students in the experimental group will be more motivated to study the material than the students in the control group. By offering innovative and interactive learning material, we believe students will be more engaged in the learning process.*

**Hypothesis 3.** *The average score of the students in the experimental group will improve more than the average score of the students in the control group in the conceptual test. We believe that by providing more engaging media, compared with just text and images, the students will retain the taught concepts better. Furthermore, the immersive nature of the Virtual Tour makes the experience closer to the in-person visits to the manufacturing plants, so that students will be able to relate the studied concepts and the real processes.*

### 4.3. Study Design

The purpose of the quantitative study approach was to identify if there were statistical differences in the conceptual assessment, motivation, and user experience between the experimental and control condition. The null hypothesis H0 was that there is no statistical difference in these measures among experimental and control conditions. The study was developed through a quasi-experimental design with pre- and post-tests. A quasi-experiment is an experiment in which participants are not assigned randomly to the control and the experimental condition [27]. Both groups, experimental and control, were groups from the course "Business production", from the Business Administration Academic program. The conditions (experimental and control) were randomized into the two groups selected for the quasi-experiment. The treatment for the experimental condition (independent variable) was the exposure to the virtual tour in a factory simulating a car seat manufacturing process, described above. The treatment for the control condition was a sketch of the factory, made by taking a screenshot from a top-view perspective of the simulation.

Sample Selection and Recruitment

The present study used data from N = 27 students of a public university in Costa Rica, 11 participants were between 18 and 25 years old, and 12 participants were between 26 and 29 years old. All participants were active students from the Business Administration academic program in this institution, specifically from the course "Business Production". To

establish the sample size, a prospective power analysis was performed with the "G*power" program [28]. To calculate the sample size in this study, an effect size of .05, two independent groups, an expected error of 0.05, and a statistical power of 0.80 were used. As a result, it was determined that a minimum of 12 participants per group was required. However, the researchers were unable to achieve the estimated sample size as the groups were defined during enrollment with a predetermined number of participants.

### 4.4. Procedure

Students from both conditions were exposed to a business production unique class where they learned about work-study techniques applied to quality processes. The learning objectives of the session for experimental and control conditions were:

- Synthesize the problems or waste that arise in the production process,
- Evaluate the production process through the use of some work-study techniques, for the reduction of waste.
- Solve through work-study tools (path diagram and analytical flowchart) manufacturing processes.

A standard protocol was used for both conditions in order to guarantee that the administration of the experiment was exactly the same for experimental and control groups, except for treatment.

Both groups completed a pre-test which included a conceptual assessment to obtain a baseline of the knowledge of students about the content selected for the study, and sociodemographic information. After the experiment, a post-test was applied to both groups, which included the conceptual assessment, the User Experience Questionnaire [29], and the Situational Motivation Scale [30].

Participants of both the experimental and control groups had to study a brief slide presentation that described the definition of work-study techniques to review quality processes, types of waste, and work-study tools (path diagram and analytical flowchart). To control both conditions, we used an active learning strategy where the recently acquired knowledge was put into practice, and both groups had to complete an analytical flowchart about the process shown in a factory. Participants in the experimental condition were exposed to the VT of the factory described above; while students from the control condition were exposed to the schematic of the plant. The schematic was produced using a top-view of the virtual tour plant and overlaying the information on top, as shown in Figure 2. The image of the workers was substituted for clarity, but the layout of the plant, the distribution of stations, and the steps of the procedure remain the same. Also, the same information is provided for both groups.

### 4.5. Measures

1. Situational Motivation Scale:

   This scale is based on the postulates of the self-determination theory of Deci and Ryan [31], which is a theoretical framework that explains the motivation within the educational setting. Situational motivation denotes a motivation that individuals experience while they engage in an activity [32]. Guay, Vallerand, and Blanchard [30] developed the Situational Motivation Scale (SIMS) to assess situational motivation; this scale comprises 16 items that assess the dimensions of intrinsic motivation, identified regulation, external regulation, and amotivation. Martín-Albo, Núñez, and Navarro [33] translated the scale into Spanish and measured its psychometric properties. The confirmatory factor analysis with a 14-item mode showed the following goodness-of-fit values for incremental fit index (IFI) = 0.93, comparative fit index (CFI) = 0.93, root mean square error of approximation (RMSEA) = 0.08, and standardized root mean square residual (SRMR) = 0.07.

   The internal consistency values of each one of the four subscales of the 14-item SIMS were 0.81 for the amotivation subscale, 0.87 for the external regulation subscale, 0.82 for the identified regulation subscale, and 0.84 for the intrinsic motivation subscale.

2. User Experience Questionnaire (UEQ):

This instrument measures the "user's perceptions and responses that result from the use and/or anticipated use of a system, product or service" (ISO 9241-210). Although a Spanish version of the UEQ was created in 2012 by Rauschenberger et al. [34], Hernández-Campos, Thomaschewski, and Law [29], they used a double-translation and reconciliation model for detecting the more appropriate words for Costa Rican culture. The scale is made up of the following six subscales:

- Attractiveness: General impression toward the product. Do users like or dislike the product? The scale is a valence dimension.
- Perspicuity: Is it easy to understand how to use the product? Is it easy to get familiar with the product?
- Efficiency: Is it possible to use the product fast and efficiently? Does the user interface look organized?
- Dependability: Does the user feel in control of the interaction? Is the interaction with the product secure and predictable?
- Stimulation: Is it interesting and exciting to use the product? Does the user feel motivated for further use of the product?
- Novelty: Is the design of the product innovative and creative? Does the product grab the user's attention?

The internal consistency values of each one of the six subscales for the Costa Rican population are attractiveness 0.87, perspicuity 0.73, efficiency 0.66, dependability 0.57, stimulation 0.69, and novelty 0.72.

3. Conceptual Assessment:

A conceptual assessment for measuring the course content was built using the specifications of the American Educational Research Association, American Psychological Association, and National Council on Measurement in Education [35], León-Velasco, Medellín Lozano, Ponce de León, and Organista-Diaz [36], and Colton and Covert [37]. The evaluation of the content was carried out through a multiple-choice test made up of 13 items. Each question had four response options, with only one correct answer. The test was aimed to evaluate students' ability to solve manufacturing processes through work-study tools. To ensure validity peer judging was performed using Fleiss's Kappa [38]. The validity of this test was guaranteed through peer judging, with almost perfect agreement ($\kappa = 0.93, 95.12\%$) [39]; however, the internal consistency of the scale was 0.384, which is a low level of reliability.

### 4.6. Qualitative Study

A focus group was conducted to further investigate the opinion of the subjects regarding the usability and contribution of the virtual tour (experimental condition) with respect to the learning process in the course [40,41]. In relation to our research question 2, we placed particular emphasis on exploring the students' perspectives regarding the immersiveness and sense of presence, if any, generated by the VT. Consequently, a significant portion of the discussions revolved around the students' perceptions of the virtual space and the realism of the presented scenes. The remaining portion of the discussion was centered on the students' perceptions of the learning experience during their utilization of the VT.

### 4.7. Sample Selection and Recruitment

The researchers used purposive sampling [41] for recruiting the participants who participated in the experimental condition. A total of 10 students participated in the focus group and provided their informed consent for participation.

### 4.8. Procedure

After implementing the experiment, a focus group was conducted with 10 students from the experimental condition to gather their opinions on the usability and effectiveness

of the virtual tour in the learning process. Prior to the start of the focus group, participants provided their informed consent to participate in this phase of the research. The discussion lasted for one hour and was recorded for transcription and analysis. The aim of this analysis was to gain an in-depth understanding of the strengths and weaknesses of the virtual tour application and learning methodology.

The data were analyzed following the principles of grounded theory. The analysis process involved two researchers and was carried out in three stages. In the first stage, each researcher individually assigned codes to relevant sentences or ideas in the transcription. In the second stage, the researchers discussed the codes they had assigned, identified common subthemes, and unified the codes. Finally, the subthemes were organized into overarching themes. Through this analysis, four themes were identified.

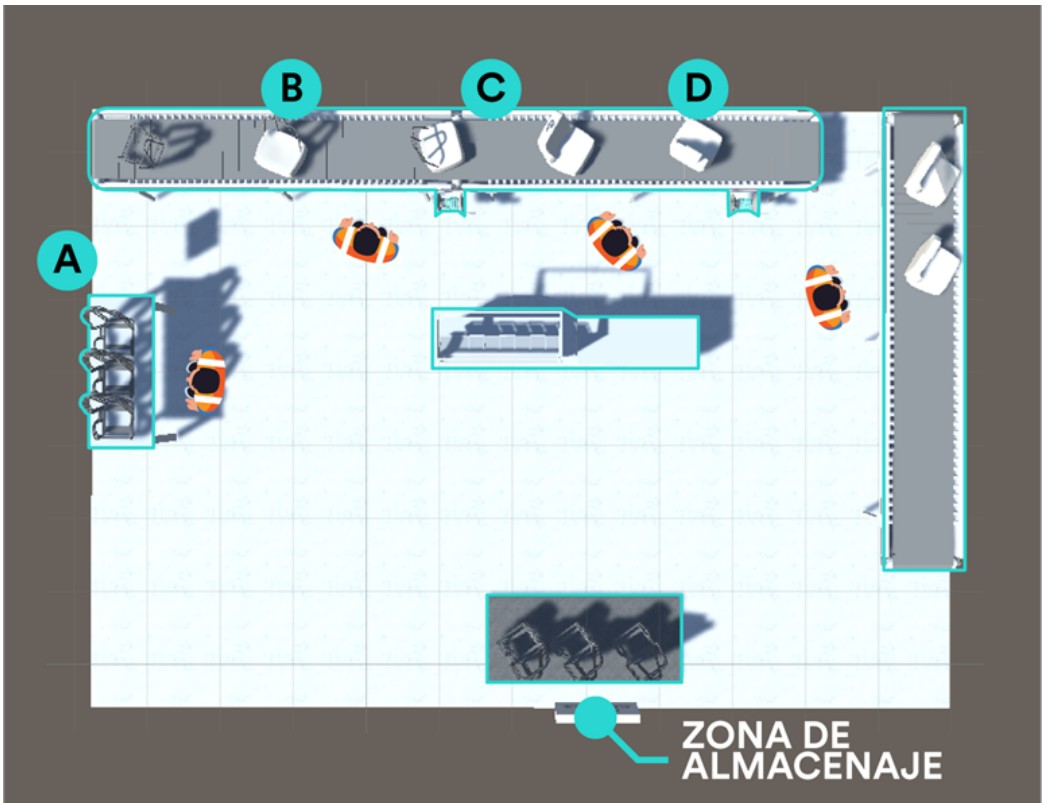

**Figure 2.** Schematic of the manufacturing plant used in the control group, produced from the top view of the virtual manufacturing plant. The stations marked A–D represent different steps in the manufaturing process, and correspond to the stations seen in Figure 1. "Zona de almacenaje" means "Storage area".

## 5. Results

### 5.1. Quantitative Results

The quantitative study was carried out using JASP version 0.16.0 [42]. A summary of the participant´s sex is described in Table 1. The descriptive statistics are presented in Table 2. Considering that the data distribution of the conceptual test, situational motivation scale, and UEQ was not normal, it was decided to use non-parametric statistics.

**Table 1.** Participant´s sex by condition.

| Condition | Sex | Frequency | Percent | Valid Percent | Cumulative Percent |
|---|---|---|---|---|---|
| experimental | male | 8 | 42.10 | 42.1 | 42.1 |
| | female | 11 | 57.8 | 57.8 | 100 |
| | Total | 19 | 100 | | |
| control | male | 3 | 37.5 | 37.5 | 37.5 |
| | female | 5 | 62.5 | 62.5 | 100 |
| | Total | 8 | 100 | | |

**Table 2.** Results of Mann–Whitney for the scales.

| | Experimental Group | | | Control Group | | | Mann–Whitney | | |
|---|---|---|---|---|---|---|---|---|---|
| | N | M | SE | N | M | SE | U | *p* | Rank-Biserial Correlation |
| Situational Motivation Scale | | | | | | | | | |
| InM | 19 | 5.27 | 1.31 | 8 | 4.09 | 1.07 | 118.50 | **0.013** | 0.559 |
| IdR | 19 | 4.09 | 1.07 | 8 | 5.12 | 1.06 | 95 | 0.162 | 0.250 |
| Exr | 19 | 4.42 | 1.06 | 8 | 4.78 | 0.76 | 58 | 0.838 | −0.237 |
| Amt | 19 | 5.77 | 1.53 | 8 | 4.53 | 1.77 | 111.50 | **0.030** | 0.467 |
| UEQ scale | | | | | | | | | |
| Attractiveness | 19 | 1.76 | 0.92 | 8 | −0.18 | 1.37 | 135.50 | **<0.001** | 0.783 |
| Perspicuity | 19 | 1.40 | 0.95 | 8 | −0.40 | 1.19 | 139.50 | **<0.001** | 0.836 |
| Efficiency | 19 | 1.61 | 1.15 | 8 | −0.18 | 1.86 | 123 | **0.007** | 0.618 |
| Dependability | 19 | 1.17 | 0.70 | 8 | −0.09 | 1.14 | 127 | **0.003** | 0.671 |
| Stimulation | 19 | 1.88 | 1.15 | 8 | 0.06 | 0.98 | 134.50 | **<0.001** | 0.770 |
| Novelty | 19 | 1.52 | 1.12 | 8 | 0.09 | 0.87 | 127 | **0.004** | 0.671 |
| Conceptual test | | | | | | | | | |
| Pre test | 19 | 5.05 | 1.54 | 8 | 5.25 | 1.83 | 70 | 0.76 | −0.079 |
| Post test | 19 | 6.47 | 1.77 | 8 | 6.37 | 2.06 | 76 | 1 | 0.000 |

InM (intrinsic motivation), IdR (identified regulation), Exr (external regulation), Amt (Amotivation). For the Mann–Whitney test, the effect size is given by the rank biserial correlation.

For testing the first hypothesis, a Mann–Whitney test was run using the subscales of the UEQ proposed by Rauschenberger et al. [34]. See Appendix B for more details. A statistically significant difference was found in all the subscales, with a higher score in the experimental group in comparison to the control group (Table 2). The effect size measure was strong for all subscales. Comparing these scores to the benchmark, we see that the virtual tour performed above average, See Figure 3. We used the provided data analysis tool which includes a consistency check using Cronbach Alpha. The alpha values for all scales were higher than the expected value of 0.7, except for Dependability. Looking closely at individual items, we found that the score for item 8 (unpredictable/predictable) was very low (0.4) which is not consistent with the scores of other items on the scale.

To test the second hypothesis, a Mann–Whitney test was conducted on the four subscales of the Situational Motivation Scale. According to the Mann–Whitney test, the experimental group achieved a significantly higher score on the subscale of intrinsic motivation (M = 5.27, SD = 1.31) compared to the control condition (M = 4.09, SD = 1.007) [U = 118.5, $p$ = 0.03]. Similar results were observed in the subscale of amotivation [U = 111.5, $p$ = 0.03], where the experimental condition exhibited a higher score (M = 5.77, SD = 1.53) compared to the control condition (M = 4.53, SD = 1.70). There were no statistically significant differences between the factors of identified regulation and external regulation. The rank biserial correlation coefficient was interpreted as a measure of effect size, which was found to be moderate for intrinsic motivation and amotivation.

To test the third hypothesis, a Mann–Whitney test was conducted on the conceptual test, but no significant differences between the groups were observed (Table 2). Additionally,

we calculated the difficulty of the test to assess concept learning. This index is defined as the quotient obtained by dividing the number of participants who correctly answer question (*A*) by the total number of participants who took the exam (*N*); its equation would be: $ID = A/N$ [43]. The equation to calculate the difficulty index of each one of the questions, correcting for the possible effects of guessing correctly, is as follows:

$$ID = \frac{A - \dfrac{E}{K-1}}{N},\tag{1}$$

where *A* is the number of participants who correctly answered the question, *E* is the number of participants who answered incorrectly, *K* is the number of alternatives to choose from, and *N* is the total number of participants. In doing so, we identified that 50% of items were very difficult (6 items), 3 items were easy, 2 items were normal, and 3 items were very easy.

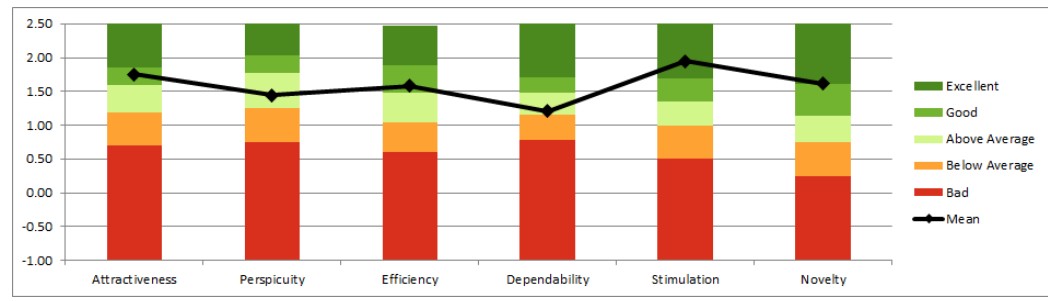

**Figure 3.** Benchmark comparison for the UEQ results (experimental group).

*5.2. Qualitative Results*

Four themes emerge from the analysis of the focus group, categorized as overall experience using the Virtual Tour Application, opinions using the application, pedagogic relevance, and other issues related to the experiment. In this section, we describe the themes and subthemes. For each subtheme, we provide an example of a comment extracted from the transcriptions and translated to English from Spanish; note that some paraphrasing might have occurred during the translation.

1. Overall experience while using the VT Application

    This theme is related to how the participants felt toward the virtual tour application in general. The comments of the study participants can be classified into two subthemes:

    (a) Motivation: participants agreed that the learning experience felt like a game, which kept them entertained and interested in the learning topic.

    *"These applications where you learn by playing and you learn while doing something, since there is more interaction, you have to be more active, there is less chance of getting bored... I liked it because I had to do something."*

    (b) Immersion: the participants mentioned that they felt as if they were in the factory. The feeling of presence, or the feeling of "being there" is an important factor in virtual reality applications.

    *"The whole environment was very good; I mean, you feel as if you are there in the factory."*

2. Opinions on the Virtual Tour Application

    Comments in this category are related to the participants' impression of the application itself, its use, and how it was presented. We identified three subthemes:

    (a) Distribution of space: this relates to how the virtual space of the factory was used, how the virtual stations were distributed, and the overall feeling of the space. This is an important subject since some of the concepts related to the topic being studied require an understanding of the distribution of space and

a correct perception of the sizes and distances between the working stations in the factory. In this regard, participants felt that the space, especially the size of objects and the distance between them was adequate.

*"I have never been in a factory, but I felt the size of the simulated one was good."*

(b) Clarity in the use of controls: for the controls we chose to use standard gaming controls (WASD for movement and mouse for viewing direction). Participants made remarks on how, although the controls were what one expects, they did not feel as smooth when compared to the controls they are used to having in video games. Participants also commented that it was not always clear what was possible to accomplish in the application, e.g., what objects are interactable and how the interaction works.

*"Have you seen at the beginning of some games, where you get a small tutorial on what each control does and how everything works? Maybe something like that would be useful."*

(c) Instructions on how to use the application: Participants argued that more detailed instructions on how to use the virtual tour were needed. During the experiment, we provided written instructions in a separate file, but the participants would have preferred that these be integrated into the virtual tour. They also suggested adding images to support the text.

*"Maybe you could present the instructions with images so that the user can better understand how to use the simulation."*

3. Learning experience

This theme relates to the overall learning experience while using the virtual tour application and how the participants felt it contributed to their learning process. We identified two subthemes:

(a) Comprehension: participants felt that the use of the virtual tour helped them to understand the studied fabrication process better since every step was shown visually. Compared to the process description they received during the test, participants felt that the one covered in the virtual tour was easier to understand. Concepts such as 'storage' and 'delay', which are related to the topic were easier to grasp, too.

*"I was able to describe the process because I could recall the mental image of each step."*

(b) Experience similar to reality: participants also commented on how the virtual tour could complement or even substitute a tour in a real factory.

*"With the COVID situation it is difficult to visit a factory, this simulation gave us an idea of how it could be."*

4. Experiment application

This theme was not originally considered when designing the focus group, but some participants made remarks on the procedure of the experiment regarding the use of the application. Some of them had difficulties downloading, installing, and running the application.

*"When I tried to run the application, I got an error 2 or 3 times and could not run it"*

## 6. Discussion

The purpose of this research was three-fold: (1) to identify if the use of VT as a pedagogic resource improves students' understanding and motivation toward the class content in comparison to the control group in a business administration academic program (2) to test if the scores of the user experience questionnaire were superior in the experimental condition in comparison to the control group, and (3) to analyze the opinion of participants of experimental condition regarding the usability and contributions of the VT.

Regarding the first research question, a Mann–Whitney analysis of the situational motivation scale revealed significant statistical differences between the control and experimental conditions for two of the subscales: intrinsic motivation and amotivation (confirming H2 of the quantitative study). According to Guay et al. [30], intrinsically motivated behaviors are those that are engaged for their own sake, in other words, for the pleasure and satisfaction derived from performing them. In this regard, students from the experimental group experienced higher intrinsic motivation while engaging in the learning activity using the VT compared to the student that used a traditional presentation, suggesting a higher level of satisfaction after performing the learning tasks. Previous research has shown that the more self-determined forms of motivation, such as intrinsic motivation or identified regulation, are more closely associated with positive consequences, such as psychological well-being and learning [30]. These results are supported by the qualitative outcomes, where participants of the experimental condition agreed that the experience using the VT kept them entertained and interested in the content of the course.

Regarding amotivation, the analysis also revealed that the participants in the control condition experienced higher levels of amotivation in comparison to participants from the experimental condition. When amotivated, individuals experience a lack of contingency between their behaviors and outcomes. Their behaviors are neither intrinsically nor extrinsically motivated. Amotivated behaviors are the least self-determined because there is no sense of purpose and no expectations of reward or possibility of changing the course of events [30]. When individuals feel a lack of autonomy, meaning they perceive their actions as controlled or imposed upon them, they may become amotivated, similarly, when individuals feel a lack of competence, they may experience amotivation. In this sense, since students in both conditions were carrying out the activities defined in the research protocol and despite participating voluntarily in this quasi-experiment, the activity did not generate any kind of expectation of reward for participation. According to the theory, the students could perceive that their actions were imposed upon them, thus, a certain level of amotivation was expected. However, this sensation was lower in the group that was able to interact with the VT. Furthermore, some authors suggest that amotivation is associated with boredom and poor concentration in educational contexts [44]. In the case of the participants from the experimental condition, during the focus group, they shared that the experience using the VT was "entertaining and interesting", which is coherent with the quantitative results. These results are consistent with Shen et al., who found that the positive emotional clues produced during immersive learning did trigger changes in the learner's mood, which can significantly enhance learners' motivation and mind-flow experience [12].

Concerning the conceptual assessment results, we did not find any significant differences in the students' performance after completing the exercises and we did not observe any improvement in the answers of either group, thus we reject hypothesis 3 of the quantitative study. Several factors could have contributed to these results. First, when we analyzed the difficulty of items, the percentage of "difficult" items was higher than recommended by literature for a written test [45]. The high level of difficulty could have influenced the results, preventing discrimination among groups. Second, although the VT and the provided material mainly focused on the manufacturing process, they did not directly relate to the concepts being tested. However, during the focus group discussions, participants expressed that the VT application helped them better understand the concepts and even the distribution of the space of the factory in the VT felt realistic. It was expected from this class that students were able to evaluate the production process through the use of some work-study techniques, for the reduction of waste. Even when it was not possible to identify differences in the final score of the test, the comprehension of distances in the space is a key element in the context of work-study techniques, this aspect was highlighted by participants as a strength of the VT. Thus, in future studies, we plan to explore the role of the VT in enhancing students' learning outcomes and evaluate its effectiveness in promoting a deeper understanding of the material.

About the third research question, participants from the experimental condition showed significantly higher scores when they evaluated the VT's user experience in comparison to the control condition evaluating the traditional material, confirming Hypothesis 1. Additionally, the VT received a high score from participants in the experimental condition compared to the benchmark. The Stimulation scale ranked in the top 10%, while Novelty and Attractiveness were classified as "Good." These findings further support the results obtained from the SIMS analysis. This was also supported by the results of the focus group regarding the immersiveness and the learning experience offered by the VT and its relation with motivation. These results can be attributed to the application of cognitive load theory principles in the design of the VT within the broader learning process. The incorporation of these principles ensured that the VT possessed suitable characteristics to effectively facilitate learning [22]. Won et al. found, in a systematic literature review, that current studies on immersive technologies in the educational field do not line up the immersive technology with a pedagogical approach [18]. In our study, the instructional design played a crucial role in shaping the VT with regard to the learning process, user experience, and cognitive load. From the outset, careful consideration was given to ensure that the VT incorporated specific characteristics aligned with these factors. Similar results using immersive technologies in the classroom have been found by Tsivitanidou et al. [13] who point out that the instructional design and the adaptation of the immersive learning tool to the learning process are crucial factors to be considered during technology integration in order to obtain positive results in learning and motivation.

Our findings have significant implications for instructional design and practice, particularly in the integration of immersive technologies within authentic educational settings. The results of our study lend support to the notion that a well-designed learning tool, which takes into account the learning process, holds the potential to enhance motivation and engagement during the learning process. Establishing a positive learning environment for students is crucial for fostering improved performance.

One limitation of our study was the relatively small sample size, which could have limited the generalizability of our findings. Furthermore, due to the quasi-experimental nature of our study conducted within a classroom setting, we were unable to randomly select individual participants. Instead, we had to randomize the assignment of entire groups to either the experimental or control group.

Additionally, as the study was conducted during the COVID-19 pandemic, students participated remotely via video calls due to health risks. This situation may have introduced uncontrolled confounding variables, such as variations in hardware setups, study environments, connectivity issues, and other distractions that could potentially impact the quality of the learning experience. Despite these limitations, we were able to perform statistical analyses on our data, and our findings offer valuable insights into the effectiveness of Virtual Tours in enhancing learning experiences. However, we acknowledge that a larger sample size would have provided more robust detection and quantification of the observed effects. Future studies with larger samples are warranted to further build upon our findings.

## 7. Conclusions

In this work, we presented an active learning methodology that integrates a VT application to help students learn about productivity concepts by showing them a step-by-step procedure to manufacture car seats. The contents of the VT were defined by the requirements derived from the learning objectives of the class, the prepared activities to be conducted during the lesson, and cognitive load theory principles. We presented the results of a mixed-design study with quantitative and qualitative data to evaluate the integration of the VT application. Through the different scales and a focus group, we show that the VT improved the students' motivation, and thus the learning experience.

Due to restrictions presented by the pandemic, the virtual tour was created as a desktop application. According to the comments obtained during the focus group discussions, this

was enough for the students to feel present in the manufacturing plant. This hints that the VT, as implemented, might be a good complement, or even substitute for in-presence visits to manufacturing plants. However, we plan to use immersive technologies, such as head-mounted displays (HMD), to enhance the experience and make it truly immersive. For future work, we plan to compare both desktop and HMD versions.

Although the subject for the Virtual Tour application was very specific to a particular lecture within an academic program, we believe that the use of immersive technologies can be used to improve the learning experience of a wider variety of domains and topics. For example, virtual environments could be used to train psycho-motor skills, e.g., manipulating laboratory equipment in a chemistry class, as has been performed in other fields [46]. Additionally, for this iteration, we focused on the learning experience, and did not investigate whether it had an influence on the overall academic results of the participants. Future work will focus on exploring these aspects.

**Author Contributions:** Conceptualization, L.C.G.-A. and R.L.-E.; Methodology, M.H.-C. and E.R.-M.; Software, E.R.-M.; Validation, J.F.A.-C.; Formal analysis, M.H.-C. and J.F.A.-C.; Investigation, Y.C.L.; Data curation, M.H.-C.; Writing—original draft, M.H.-C. and Y.C.L.; Writing—review and editing, L.C.G.-A., J.F.A.-C., E.R.-M. and R.L.-E.; Visualization, L.C.G.-A.; Supervision, R.L.-E.; Project administration, Y.C.L. All authors have read and agreed to the published version of the manuscript.

**Funding:** This research received no external funding.

**Informed Consent Statement:** Informed consent was obtained from all subjects involved in the study.

**Data Availability Statement:** Data can be provided by request.

**Conflicts of Interest:** The authors declare no conflict of interest.

## Abbreviations

The following abbreviations are used in this manuscript:

| VT | Virtual Tour |
|----|-------------|
| RQ | Research Question |
| SIMS | Situational Motivation Scale |
| UEQ | User Experience Questionnaire |

## Appendix A

A confirmatory factor analysis was carried out in order to corroborate that the data obtained with the Situational Motivation Scale (SIMS) were grouped according to Martín-Albo, Núñez, and Navarro [33]. According to the chi square, the model does not have a good fit; however, the literature has shown that the chi square is sensitive to small sample sizes and non-normal samples. Regarding to the Ratio X2, the model does present a good fit. The similar structure fits well with the theoretical perspective, and therefore subsequent analyzes was carried out using the derived factors.

**Table A1.** Model fit.

| Chi2 | Df | *p* | Ratio X2/ DF |
|------|-----|-----|--------------|
| 188.834 | 98 | <0.001 | 1.92 |

According to the composite reliability, all factors have high reliability. So, the factorial structure proposed in the original study is adequate to make the independent samples *t*-test between experimental and control group. In addition, all factor loadings are superior to 0.4.

**Table A2.** Factorial weights, explained variance and reliability of each factor.

| Factor | Item | Factor Loading | Variance Explained | Composite Reliability |
|---|---|---|---|---|
| Intrinsic motivation | 1 | 0.725 | 66.64% | 0.87 |
| | 5 | 0.952 | | |
| | 9 | 0.800 | | |
| | 13 | 0.768 | | |
| Identified regulation | 2 | 0.964 | 59.95% | 0.824 |
| | 6 | 0.907 | | |
| | 10 | 0.252 | | |
| | 14 | 0.961 | | |
| External regulation | 3 | 0.864 | 59.37% | 0.851 |
| | 7 | 0.927 | | |
| | 15 | 0.853 | | |
| | 11 | 0.794 | | |
| Amotivation | 4 | 0.972 | 60.74% | 0.854 |
| | 8 | 0.882 | | |
| | 12 | 0.932 | | |
| | 16 | 0.964 | | |

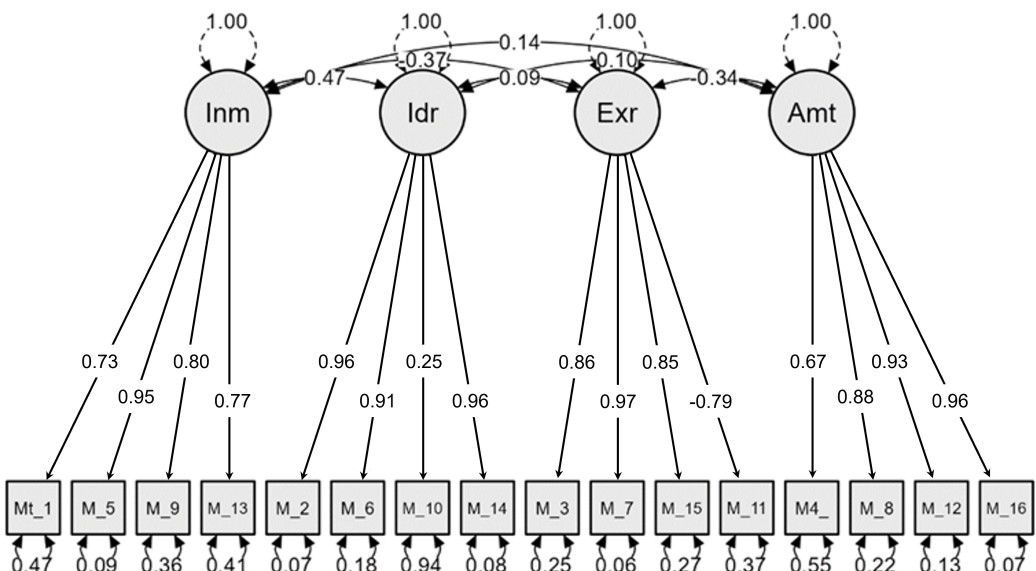

**Figure A1.** Confirmatory factor analysis model plot.

**Appendix B**

The researchers of this study had the purpose of validating the factorial structure of the UEQ presented by Schrepp et al. [47]. However, it was not possible to carry out this task because confirmatory factor analysis (CFA) requires the number of observations be greater than the number of variables, and this parameter was not met due to the small sample size. For this reason, subsequent tests were performed using the original factorial structure proposed by the Schrepp et al. [47].

The internal consistency of each subscale was high despite of the sample size: attractiveness 0.94, perspicuity 0.83, efficiency 0.94, dependability 0.69, stimulation 0.80, and novelty 0.82.

## Appendix C

**Table A3.** Difficulty analysis of the items of the conceptual test.

| Item | A | N | E | K | ID | Corrected ID | Classification |
|------|-----|-----|-----|-----|------|--------------|----------------|
| 1 | 19 | 27 | 8 | 4 | 0.70 | 0.60 | easy |
| 2 | 5 | 27 | 22 | 4 | 0.19 | −0.09 | very hard |
| 3 | 16 | 27 | 11 | 4 | 0.59 | 0.46 | normal |
| 4 | 10 | 27 | 17 | 4 | 0.37 | 0.16 | very hard |
| 5 | 4 | 27 | 23 | 4 | 0.15 | −0.14 | very hard |
| 6 | 18 | 27 | 9 | 4 | 0.67 | 0.56 | easy |
| 7 | 24 | 27 | 3 | 4 | 0.89 | 0.85 | very easy |
| 8 | 22 | 27 | 5 | 4 | 0.81 | 0.75 | very easy |
| 9 | 16 | 27 | 11 | 4 | 0.59 | 0.46 | normal |
| 10 | 5 | 27 | 22 | 4 | 0.19 | −0.09 | very hard |
| 11 | 9 | 27 | 18 | 4 | 0.33 | 0.11 | very hard |
| 12 | 24 | 27 | 3 | 4 | 0.89 | 0.85 | very easy |
| 13 | 2 | 27 | 25 | 4 | 0.07 | −0.23 | very hard |

## Appendix D

**Table A4.** Association between tools, types of assets, format and the development phase of the Virtual Tour Application in which they were used.

| Tool | Type | Asset Type | Format | Phase 1 | Ctrl 2 | Exp |
|------|------|------------|--------|---------|--------|-----|
| Sketchfab | Online plat | Share Link | URL | x | | |
| SketchUpPro Trimble | Application | 3D models | OBJ/FBX | x | | x |
| Adobe Illustrator | Application | Vector | SVG | | x | x |
| Adobe Audition | Application | Voice-over audio | MP3 | x | | x |
| Adobe After Effects | Application | Animation sequence | PNG | | | x |
| TexturePacker | Application | Multi-Spritesheet 2D animations | PNG | | | x |
| Unity | Application | Web App | WebGL | | | x |
| Blender | Application | 3D models | OBJ/FBX | | | x |
| MS Power Point | Application | SlideShow Presentation | PPTX | | x | |

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
