# Peer review of "Improving Motivation and Learning Experience with a Virtual Tour of an Assembly Line to Learn about Productivity"

_sustainability, doi:10.3390/su151411407_

Round 1

Reviewer 1 Report

It is commendable that with such a small size, the authors managed to do such a rigourus statistical analysis to test the hypotheses  

Paper need proof reading, it has typos. For example, line 220 reads "aas" instead of as. The description of what was being compared is not very explicit. What did the control group experience that was being compared to VT? Not very clear what the control group experienced that was being compared. If they did not experience any form of tour, then how is it different from just comparing motivation before and after?

Since VT is a cognitive domain, it would be interesting on how that translates into psychomotor... does that help students learn better after the VT?

Need some proof reading

Reviewer 2 Report

Dear authors,

An excellent work, interesting and well-argued from a scientific point of view. A very consistent methodology and well-chosen statistical methods.

It could be a model for analyzing any new educational approach in a specific field of learning. 

It is recommended an analysis of differences between males and females, knowing that there are differences in their perceptual characteristics and attitude versus technology. 

It could be also interesting to see if the academic results of the students were improved. Even the motivation was increased, there are also other factors influencing the learning results. Some comments on this direction should be added.

A limit of the research or a future direction of development should be mentioned: the transfer of knowledge achieved in the VT environment into practice. 

Some observations:

A correction in table 2: InM or Int?

Line 220, it needs a correction. 

Good luck!
